# Topography and motion of acid-sensing ion channel intracellular domains

Tyler Couch[1], Kyle D Berger[2], Dana L Kneisley[2], Tyler W McCullock[1], Paul Kammermeier[2], David M Maclean[2]*

[1]Graduate Program in Cellular and Molecular Pharmacology and Physiology, Reno, United States; [2]Department of Pharmacology and Physiology, University of Rochester Medical Center, New York, United States

**Abstract** Acid-sensing ion channels (ASICs) are trimeric cation-selective channels activated by decreases in extracellular pH. The intracellular N and C terminal tails of ASIC1 influence channel gating, trafficking, and signaling in ischemic cell death. Despite several X-ray and cryo-EM structures of the extracellular and transmembrane segments of ASIC1, these important intracellular tails remain unresolved. Here, we describe the coarse topography of the chicken ASIC1 intracellular domains determined by fluorescence resonance energy transfer (FRET), measured using either fluorescent lifetime imaging or patch clamp fluorometry. We find the C terminal tail projects into the cytosol by approximately 35 Å and that the N and C tails from the same subunits are closer than adjacent subunits. Using pH-insensitive fluorescent proteins, we fail to detect any relative movement between the N and C tails upon extracellular acidification but do observe axial motions of the membrane proximal segments toward the plasma membrane. Taken together, our study furnishes a coarse topographic map of the ASIC intracellular domains while providing directionality and context to intracellular conformational changes induced by extracellular acidification.

*For correspondence:
David_MacLean@urmc.rochester.edu

Competing interests: The authors declare that no competing interests exist.

## Introduction

The predominant sensor of extracellular pH in the central and peripheral nervous system is thought to be acid-sensing ion channels (ASICs) (*Pattison et al., 2019*). This family of trimeric pH-activated sodium-selective ion channels couples extracellular acidification to numerous physiological and pathophysiological consequences including synaptic transmission and plasticity (*Du et al., 2014*; *Kreple et al., 2014*; *Liu et al., 2016*; *Uchitel et al., 2019*; *Yu et al., 2018*), fear memory consolidation and expression (*Du et al., 2017*; *Taugher et al., 2020*; *Taugher et al., 2017*; *Taugher et al., 2014*), epilepsy (*Ziemann et al., 2008*), and ischemia-induced cell death (*Wemmie et al., 2013*; *Xiong et al., 2004*). However, in many of these cases, the duration of extracellular acidosis is quite long. Since most ASIC subunits desensitize nearly completely in a few seconds (*Benson et al., 2002*; *Gründer and Pusch, 2015*), how might these channels convey information over minutes or hours? While other extracellular pH sensors may play roles (*Pattison et al., 2019*; *Ruan et al., 2020*), recent evidence reveals that ASICs possess both canonical ionic and non-canonical metabotropic signaling capabilities.

Extracellular acidosis is proposed to drive the association of the serine/threonine kinase receptor interacting protein 1 (RIP1) and *N*-ethylmaleimide-sensitive factor with ASIC1a, triggering RIP1 phosphorylation and initiating a necroptotic signaling cascade (*Wang et al., 2020*; *Wang et al., 2015*). This occurs independent of ASIC1a ion conductance and via intracellular protein-protein interactions (*Wang et al., 2020*; *Wang et al., 2015*). Thus, ASICs join a growing, and controversial, list of ligand-gated ion channels who also possess metabotropic signaling capacity (*Dore et al., 2017*; *Kabbani and Nichols, 2018*; *Pressey and Woodin, 2021*; *Rodríguez-Moreno and Sihra, 2007*; *Valbuena and Lerma, 2016*). Like many of these ion channels, the structure of the ASIC extracellular

and transmembrane regions has been determined (*Baconguis et al., 2014*; *Gonzales et al., 2009*; *Yoder and Gouaux, 2020*; *Yoder et al., 2018*), but the intracellular domains (ICDs) are unresolved. It is likely that the ASIC ICDs are partly, or even mostly, unstructured regions. Nonetheless, understanding the coarse outline of the two ICDs, the N and C tails, as well as their pH-induced movements, would inform further investigation of ASIC metabotropic signaling and may shed light on how the ICDs impact gating, protein-protein interactions, and/or trafficking (*Baron et al., 2002*; *Chai et al., 2007*; *Hruska-Hageman et al., 2004*; *Kellenberger and Schild, 2015*; *Klipp et al., 2020*; *Leonard et al., 2003*; *Schnizler et al., 2009*).

Here, we set out to generate a coarse topography of the chicken ASIC1 intracellular N and C termini using fluorescence resonance energy transfer (FRET) in several configurations. We find that the short (15 amino acid) N tail projects about 6–10 Å from the membrane inner leaflet into the cytosol while the longer 67 amino acid C tail extends between 30 and 40 Å. This contrasts with the unresolved tail of AMPA receptors that is largely perpendicular to the membrane (*Zachariassen et al., 2016*). We also report that the N and proximal segment of the C tail from the same subunits are closer than adjacent subunits, suggesting preferential intra-subunit tail interactions which may inform the interpretation of experiments with heteromeric channels. Finally, using pH-insensitive fluorescent proteins (FPs), we fail to detect any relative movement between the N and C tails but do observe axial motions of these segments toward the plasma membrane upon extracellular acidification. Together, these data allow us to build a topographical model of the intracellular tails of ASIC1 that will be a foundation for working hypotheses in future experiments.

## Results

### Voltage-dependent quenching measurements with dipicrylamine

The intracellular components of the cASIC1 N and C terminal tails are 15 and 67 amino acids, respectively. The range of possible structures adoptable by these tails is quite large. For example, in principle, the C tail could exist as a linear strand of peptide more than 70 Å long (*Miller and Goebel, 1968*) extended parallel to the plasma membrane or as a similar extended strand but perpendicular to the membrane plane, projecting 'downward' into the cytosol. These two extreme scenarios represent the borders of the conformational landscape. The main goal of our study was to experimentally shrink the range of possibilities and map this landscape. Our secondary goal was to determine if these ICDs move upon extracellular acidification. To do this, we adopted a FRET approach using different colored FPs as donors and the membrane soluble small molecule DPA as an acceptor (*Figure 1A*). DPA has several advantageous properties as a FRET acceptor (*Chanda et al., 2005b*; *Wang et al., 2010*). First, it is a dark acceptor, allowing FRET to be easily measured by changes in donor emission. Second, it binds and dissociates from the plasma membrane quickly with respect to the time of a patch clamp experiment. Third, and most critically, the anionic DPA orients in the plasma membrane depending on voltage, being pulled into the inner membrane leaflet by depolarization or pushed to the outer leaflet by hyperpolarization. Thus, voltage steps can move DPA molecules closer to, or further from, the donor by a fixed distance of 25 Å, reflecting to the estimated displacement of DPA (*Wang et al., 2010*; *Zachariassen et al., 2016*). These qualities have been used to probe topography and conformational changes in several ion channels from multiple groups (*Barros et al., 2018*; *Chanda et al., 2005a*; *De-la-Rosa et al., 2013*; *Groulx et al., 2010*; *Taraska and Zagotta, 2007*). We validated this approach using patch clamp fluorometry where the fluorescence from single patch clamped cells, transfected with membrane tethered CFP or YFP variants (mTurquiose2 and mVenus), was recorded in the continual presence of DPA during a series of voltage steps (−180 to +120 mV, 20 mV increments, *Figure 1B*). Membrane tethered CFP (Lck-CFP) gave robust voltage-dependent fluorescence changes in the presence of DPA (5 µM) but not in its absence and was quantified as the difference between the top and bottom values of a Boltzmann sigmoidal fit of the normalized fluorescence ($\Delta F/F_{-15\ mv}$ = 0.45 ± 0.03, $n$ = 6, *Figure 1C & D*), consistent with the fluorophore being close to the plasma membrane and having a relatively high $R_0$ distance (47 Å). The Lck-YFP gave relatively weak voltage-dependent quenching in the presence of DPA ($\Delta F/F_{-15\ mv}$ = 0.16 ± 0.013, $n$ = 6, *Figure 1C & D*), which is also consistent with a membrane proximal position but relatively lower $R_0$ distance (36.5 Å).

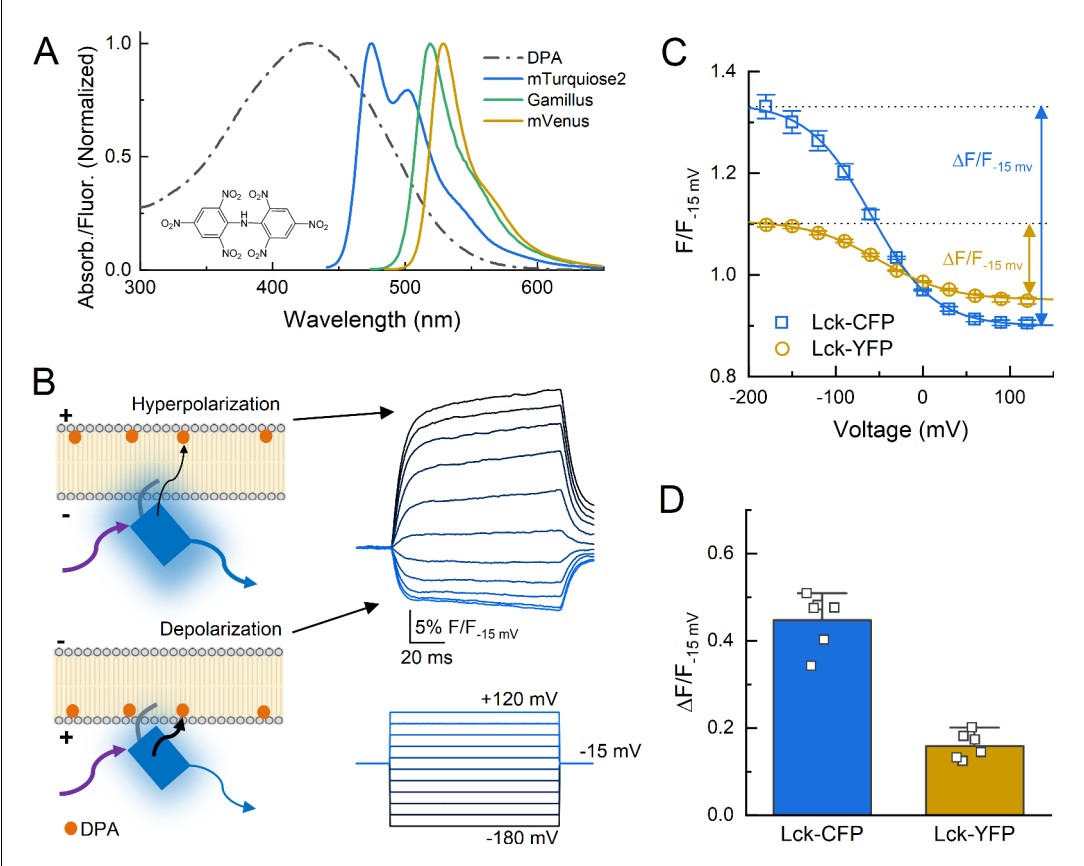

**Figure 1.** Schematic of dipicrylamine (DPA) quenching measurements. (**A**) Absorption (*dotted line*) or fluorescence emission (*solid lines*) spectra of DPA (*black*), CFP variant mTurqoise2 (*blue*), GFP variant Gamillus (*green*) and YFP variant mVenus (*yellow*). Structure of DPA inset. (**B**, *left*) DPA (*orange circle*) localizes to the outer leaflet with hyperpolarization and moves to inner leaflet upon depolarization, resulting in stronger quenching. Arrows reflect excitation (*purple*), emission (*blue*), or resonance transfer (*black*). (**B**, *right*) Exemplar patch fluorometry data of Lck-CFP fluorescence changing during voltage steps. (**C**) Single cell fluorescence, normalized to fluorescence at −15 mV, over a range of voltages for Lck-CFP (blue) and Lck-YFP (*yellow*). Solid lines are fits to Boltzmann function which yield Δ*F/F*. (**D**) Summary of Δ*F/F*s from Lck-CFP or Lck-YFP. Squares are single cells (*N* = 6 cells per construct) and error bars depict SEM.

The online version of this article includes the following figure supplement(s) for figure 1:

**Figure supplement 1.** Non-linearity of fluorescence resonance energy transfer (FRET) provides for distance estimates.

Next, we set out to generate a coarse topographical map of chicken ASIC1 using DPA quenching. To do this, we relied upon the steeply non-linear distance dependence of FRET. For example, if DPA is close to a donor fluorophore then inducing a 25 Å displacement away using voltage steps will produce a large change in fluorescence whereas if DPA is far from a donor then a 25 Å displacement away will yield a much smaller effect, provided the DPA-donor initial distances are not on the asymptotes of the FRET curve (*Figure 1—figure supplement 1*). We inserted CFP or YFP immediately upstream of the N terminus and into four positions within the C tail (*Figure 2A*, *Figure 2—figure supplement 1*). The C tail positions were selected based on prior divisions within this domain (*Wang et al., 2020*; *Wang et al., 2015*). If the C tail is roughly parallel to the plasma membrane, as observed for AMPA receptors (*Zachariassen et al., 2016*), then all insertions will have equal voltage-dependent quenching. However, if quenching varies at different insertion points and between fluorophores, these data will provide insight into the topography of the tail. Importantly, FP insertions did not disrupt channel function as determined by the rates of desensitization and recovery, although small differences in desensitization decay of C4 and recovery of C1 insertions appeared (*Figure 2—figure supplement 2*). Further, these constructs showed robust plasma membrane localization in confocal microscopy (*Figure 2—figure supplement 3*). For each construct, we measured the fluorescence emission from single voltage clamped cells over a range of voltage steps before

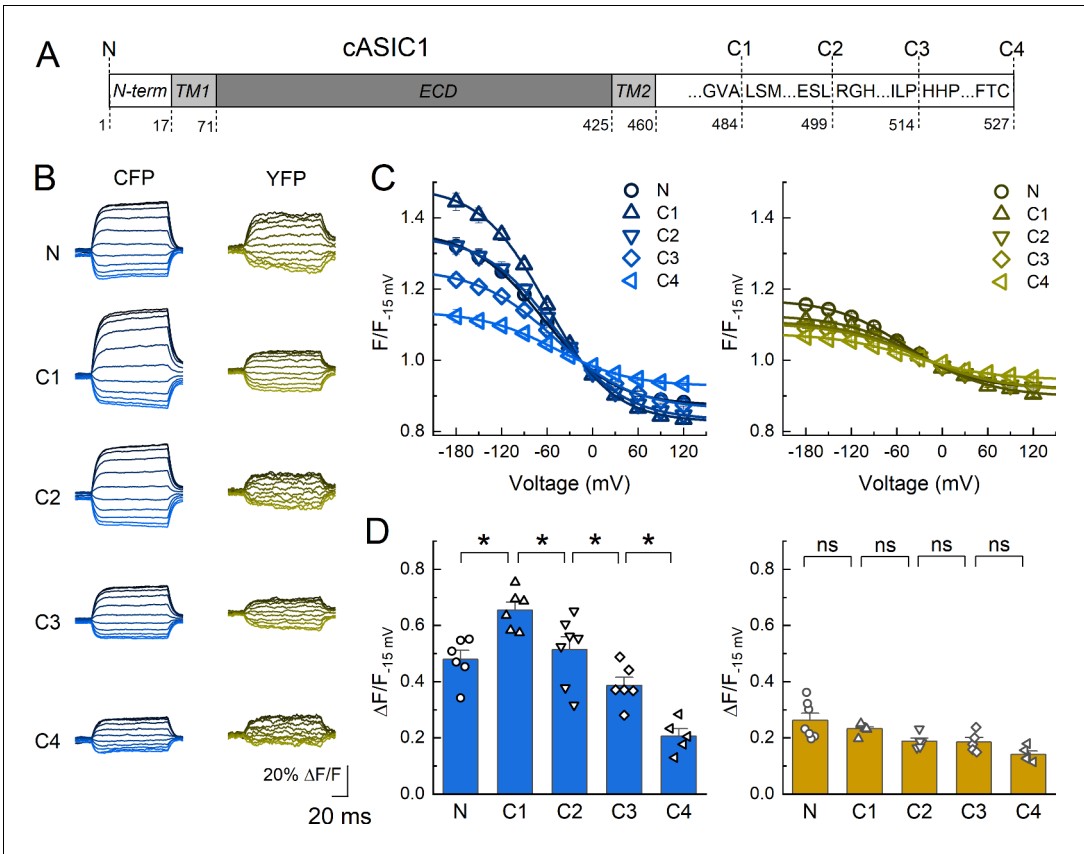

**Figure 2.** Dipicrylamine (DPA) quenching of CFP or YFP insertions in cASIC1 intracellular domains. (**A**) Schematic of cASIC1 constructs used. The extracellular domain (*ECD*) and transmembrane helices (*TM*) are gray with amino and carboxy termini depicted in white. Letters within the carboxy tail indicate the amino acid sequence around the insertion site while the lower numbers give the position. (**B**) Example fluorescence traces for each of the CFP or YFP insertions. Voltage protocol and coloring is the same as in *Figure 1B*. (**C** and **D**) Quenching curves and summary plots for CFP (*left*) and YFP (*right*) insertions. Symbols denote individual cells (*N* = 5–7 cells) and error bars represent SEM. Asterisks mark significant differences with p values of 0.004 (N–C1), 0.0475 (C1–C2), 0.0483 (C2–C3), and 0.0163 (C3–C4) for CFP.

The online version of this article includes the following figure supplement(s) for figure 2:

**Figure supplement 1.** cASIC1 constructs used in this study.

**Figure supplement 2.** Fluorescent protein (FP) insertion does not substantially alter desensitization kinetics.

**Figure supplement 3.** Lck-CFP and CFP-tagged cASIC1 constructs localize to the plasma membrane.

**Figure supplement 4.** Dipicrylamine (DPA) quenching of CFP or YFP insertions in cASIC1 intracellular domains without Boltzmann fits.

**Figure supplement 5.** Theoretical analysis of fluorescent protein-dipicrylamine (FP-DPA) quenching.

**Figure supplement 6.** Distance in extracellular domain for reference.

DPA application, during continual application of 5 µM DPA and during a subsequent wash period. *Figure 2* shows the results of these experiments with extracellular pH 8 to maximally populate the resting state. Importantly, the extent of quenching varied by position, ruling out the possibility that the C tail is parallel to the membrane. Specifically, we found that with CFP insertions, the C1 position gave the strongest quenching ($\Delta F/F_{-15\ mV} = 0.65 \pm 0.03$) with N and C2 being about equal ($\Delta F/F_{-15\ mV} = 0.48 \pm 0.03$ and $\Delta F/F_{-15\ mV} = 0.51 \pm 0.05$) and C3 and C4 having progressively less quenching ($\Delta F/F_{-15\ mV} = 0.39 \pm 0.03$ and $\Delta F/F_{-15\ mV} = 0.21 \pm 0.03$). In the case of YFP, a slightly different pattern was observed where the N terminal YFP gave the strongest quenching ($\Delta F/F_{-15\ mV} = 0.26 \pm 0.02$) with C1–C4 giving progressively weaker quenching ($\Delta F/F_{-15\ mV}$ between $0.23 \pm 0.02$ and $0.14 \pm 0.01$, *Figure 2B–D*). Quenching of the N terminal YFP insertion was statistically distinguishable

from the C4 insertion (p = 0.019) but did not detect differences between adjacent insertions as for CFP (*Figure 2D*). As with the Lck data, we used Boltzmann fits to define the maximum and minimum fluorescence and calculate a $\Delta F/F$ (see Materials and methods and *Figure 1*). However, using a 'fit free' approach and calculating a $\Delta F/F$ by subtracting the measured fluorescence at the extreme voltages (i.e. −180 mV and +120 mV) divided by the fluorescence at −15 mV produces similar results with an identical pattern of statistical significance (*Figure 2—figure supplement 4*).

To gain further insight from these data, we modeled DPA quenching of CFP and YFP using previously established theory (*Wang et al., 2010*; *Zachariassen et al., 2016*; *Figure 2—figure supplement 5*). The expected quenching of CFP and YFP was simulated at infinitely negative and positive voltages over a range of distances from the plasma membrane (*Figure 2—figure supplement 5A and B*). Subtracting these curves and normalizing produces the theoretical $\Delta F/F_{norm}$ curves as a function of distance for both CFP and YFP (*Figure 2—figure supplement 5C*). The CFP curve has a distinct hump at 24 Å (i.e. 20 and 28 Å give equal extents of quenching) while the YFP curve peaks at 16 Å and falls off with increasing distance (*Figure 2—figure supplement 5C*). This broadly mirrors the quenching pattern we observe in our data with CFP C1 giving the greatest quenching followed by N and C2 then C3 and C4 while in YFP all positions are weaker than N (*Figure 2—figure supplement 5C*). The most parsimonious interpretation of these data is that the N terminal insertion point is membrane proximal while all positions in the C tail are progressively further from the membrane. To assign some provisional distances to these measurements, we assumed the FP adds 10 Å to the axial distance, reflecting the radius of the beta barrel. With this assumption, the strongest interpretation of our data is that the tip of the N terminus is about 6–10 Å from the inner leaflet while the C1 insertion site is approximately 12–16 Å away. Distance estimates for the C2–C4 insertion sites are difficult to assign; however, given that the C4 position shows quenching for both FPs, this final C terminal amino acids are likely within 30–40 Å of the inner leaflet. For reference, the C-α to C-α distance from the critical His74 to Lys355, at the base of thumb domain's alpha5 helix, is 35 Å (*Figure 2—figure supplement 6*, resting state, PDB:6VTL). Uncertainties and caveats to these estimates are enumerated in the Discussion section. Increases in accuracy to this provisional map will require reductions in background fluorescence and a smaller fluorophore at more positions.

## N and C tail inter- versus intra-subunit interactions

The DPA quenching experiments indicate the N terminal insertion is closest to the plasma membrane, while the C1–C4 insertions are progressively further away. Within the membrane proximal segment of the C tail is a cluster of lysine and arginine residues (amino acids 461–476). These have been proposed to interact with a segment of negatively charged amino acids in the N termini (amino acids 7–12), forming a salt bridge between the intracellular tails of the same subunit (*Wang et al., 2020*; *Wang et al., 2015*). We reasoned that if such a bridge was formed between tails of the same subunit, then FPs appended to both tails of the same subunit would experience stronger FRET than FPs on the tails of different subunits. To test this, we used fluorescent lifetime imaging (FLIM) to measure the lifetime of an N terminal CFP donor in the absence or presence of a C terminal YFP acceptor. A challenge in this experiment is ensuring that only one FRET pair exists within the majority of trimers. If more than one pair of FPs are present, a 'donor-centric' measure of FRET like FLIM will be skewed toward higher FRET efficiencies (*Ben-Johny et al., 2016*). To measure the FRET between CFP and YFP in a single subunit, we appended a C terminal YFP to the N terminal CFP clone (CFP-cA1-YFP, *Figure 2—figure supplement 1*). This construct was transfected along with wild-type cASIC1 at progressively increasing ratios. In principle, as the amount of unlabeled wild-type cASIC1 increases, the FRET should decrease (lifetimes increase) until a plateau is reached. The FRET value of this plateau will correspond to the FRET from single subunits. As expected, CFP-cA1-YFP alone produced very strong FRET as evidenced by the dramatically faster CFP lifetimes compared to N CFP (N CFP: 4.06 ± 0.02 ns, n = 76; CFP-cA1-YFP: 2.68 ± 0.03, n = 58, *Figure 3*). Adding more wild-type cASIC1 led to an increase in CFP lifetimes until reaching a plateau of approximately 3 ns, representing a FRET efficiency of 25% between CFP and YFP in a single subunit (1:31 ratio: 3.08 ± 0.03, n = 34, 24% FRET efficiency, *Figure 3C*).

To measure the FRET between FPs in different subunits, we concatenated two cASIC1 subunits together in a dimer with an N terminal CFP (CFP-cA1-cA1, *Figure 2—figure supplement 1*). Expression of this clone along with wild-type cASIC1 gave the expected approximately 4 ns mono-exponential lifetime from this CFP variant (3.86 ± 0.02 ns, n = 74). Co-transfection of CFP-cA1-cA1 along

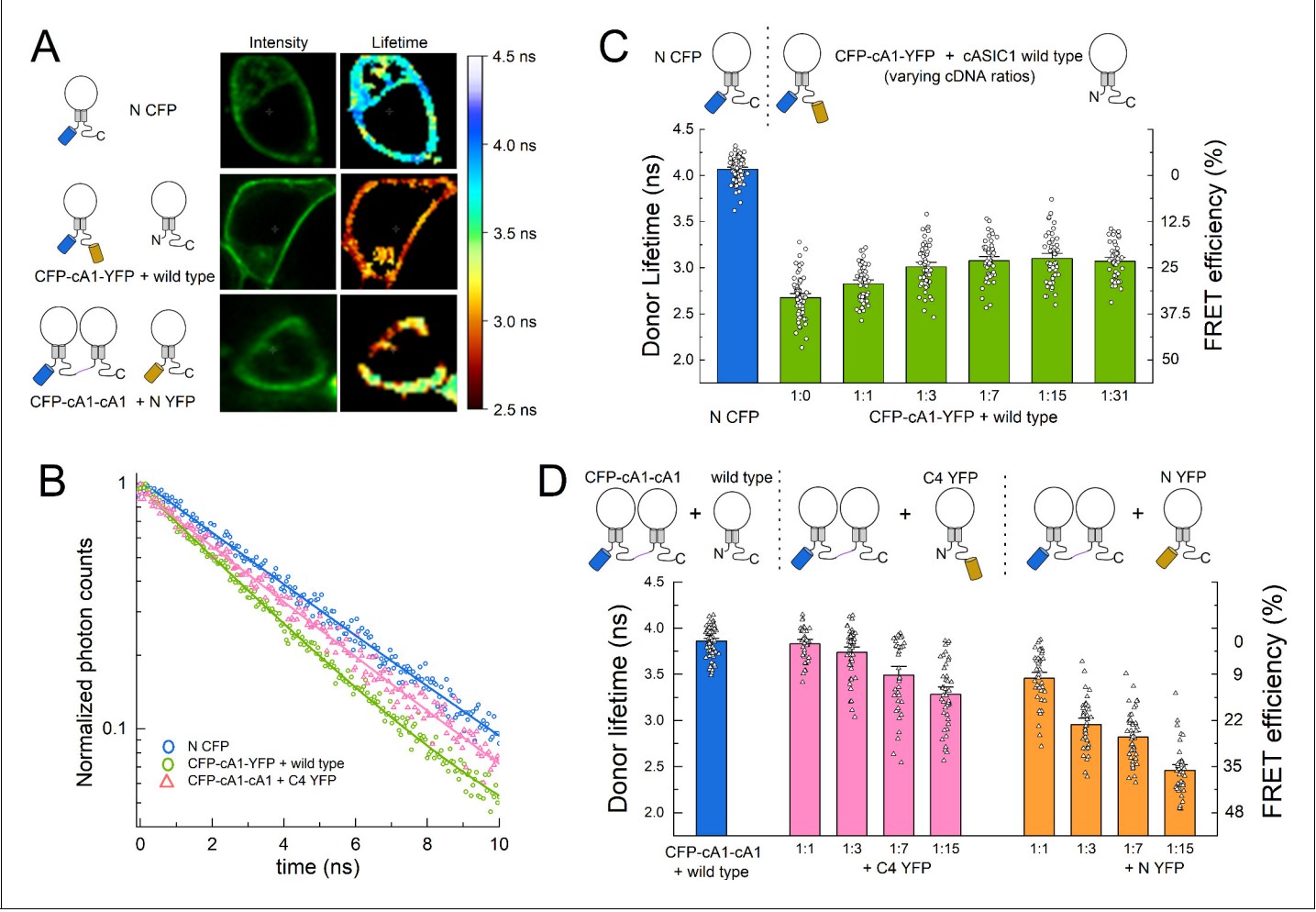

**Figure 3.** Fluorescence resonance energy transfer (FRET) is stronger within subunits than between subunits. (A) Example cells transfected with the indicated constructs showing intensity (*left*) and lifetime (*right*) images. (B) Time-correlated single photon counting (TCSPC) histograms from the plasma membrane of cells in A and additional control. Solid line is a single exponential fit to the data. (C) Cartoon (*upper*) and lifetimes (*lower*) from the indicated constructs for measuring intra-subunit FRET. (D) Cartoon (*upper*) and lifetimes (*lower*) for measuring inter-subunit FRET. Symbols denote single cells (*N* = 35–76 cells) and error bars are SEM.

The online version of this article includes the following figure supplement(s) for figure 3:

**Figure supplement 1.** Fluorescence resonance energy transfer (FRET) due to membrane 'crowding'.

with C4 YFP at increasing ratios produced progressively faster lifetimes up to about 3.3 ns or 15% FRET efficiency (1:15: 3.28 ± 0.05 ns, *n* = 43, *Figure 3*). In this experiment, FRET comes partly from the CFP-cA1-cA1 transferring energy to a YFP within the channel complex and partly from transfer to YFPs in nearby ASICs (i.e. crowding). This crowding effect is illustrated in *Figure 3—figure supplement 1* where CFP-cA1-cA1 is co-expressed with unlabeled wild type and Lck-YFP at varying ratios. As the Lck-YFP amount is increased so does the FRET efficiency. Separating out the fraction of FRET due to crowding is difficult but also unnecessary in this case. Even with a 1:15 ratio of CFP-cA1-cA1 to C4 YFP, where all dimers likely have a C4 YFP monomer partner, the FRET efficiency is significantly less than that observed in the CFP-cA1-YFP experiment (15 ± 1%, *n* = 43 for CFP-cA1-cA1 + C4 YFP 1:15; 24 ± 1%, *n* = 46 for CFP-cA1-YFP + wild type 1:15, p < 1e−6, Student's t-test). Thus, FPs on the same subunit show more efficient FRET than on different subunits, suggesting the tails of the same subunit are physically closer and more likely to associate.

The addition of C4 YFP at 1:1 and 1:3 ratios gave unexpectedly small FRET efficiencies (1 ± 1% and 3 ± 1%, *n* = 35 and 46). To determine if the concatenated dimer was assembling as anticipated, we repeated the experiment using N YFP, where the acceptor is positioned much closer to the

donor and hence should give more robust FRET. Consistent with this, all ratios of N YFP gave a strong reduction in donor lifetimes, indicating the CFP dimer does assemble with labeled monomer. Taken together, these experiments demonstrate that FRET efficiency from FPs on neighboring subunits is considerably less than that measured from the same subunit, supporting a model where the N and C tails of the same subunit interact.

## Measurement of ICD conformational changes

The putative salt bridge between the N terminal tail and the membrane proximal segment of the C tail is proposed to break apart to facilitate ion-independent signaling (*Wang et al., 2020*). Evidence for this movement comes from the change in FRET between N terminal CFP and C terminal YFP upon extracellular acidification (*Wang et al., 2020*). To investigate this, we conducted an analogous experiment using patch clamp fluorometry with simultaneous recording of both the blue and yellow emission from CFP-cA1-YFP receptors. As seen in *Figure 4A and B*, rapid application of pH 7 or 6, but not pH 8, produced a decrease in the ratio of YFP to CFP emission (i.e. change in FRET) with a slow time course ($\Delta F_{\text{YFP/CFP}}$ pH 7: $-2.6 \pm 0.3\%$, $n = 8$, $p = 0.031$ versus pH 8; pH 6: $-4.6 \pm 0.5\%$, $n = 8$, $p = 0.0025$ versus pH 8). However, as discussed previously (*Wang et al., 2020*), the strong pH sensitivity of YFP complicates the interpretation of these measurements. Consistent with this (*Wang et al., 2020*), we also observed a strong loss of emission from C4 YFP upon extracellular acidification (*Figure 4A and B*, $\Delta F_{\text{YFP}}$ pH 7: $-2.6 \pm 0.8\%$, $n = 5$; pH 6: $-12 \pm 2\%$, $n = 8$). In contrast, the N CFP showed no such change in emission (*Figure 4*) owing to this CFP variants pKa of 3.1 (*Goedhart et al., 2012*).

Blue/yellow and green/red pairings are the most common for FRET. The chromophores of GFP and YFP are intrinsically pH sensitive making them unsuitable for studying acid-evoked conformational changes. Fortunately, a novel GFP was described with a distinct chromophore and pKa of 3.4 (*Shinoda et al., 2018*). When combined with a relatively low pKa red FP (TagRFP, pKa 3.8 *Merzlyak et al., 2007*), this pairing should enable a FRET measurement that is relatively insensitive to the pH range examined while gaining the advantages of green/red FRET (*McCullock et al., 2020*). We therefore cloned the low pKa GFP into the N terminal position and the RFP into the C4 position (GFP-cA1 and cA1-RFP, *Figure 2—figure supplement 1*) and tested the pH sensitivity of their fluorescence emission. As seen in *Figure 4C and D*, both clones showed no change in their fluorescence emission with extracellular pH 6, 7, or 8, confirming their suitability for measuring conformational changes induced by pH. We next added both FPs into a single subunit (GFP-cA1-RFP) and repeated our dual-channel patch clamp fluorometry measurements. However, we detected no change in the RFP/GFP ratio (i.e. no change in FRET) from these FPs (*Figure 4C and D*, $\Delta F_{\text{RFP/GFP}}$ pH 7: $-1.3 \pm 0.5\%$; pH 6: $-0.1 \pm 0.6\%$, $n = 5$). Importantly, the GFP does FRET with the RFP as the ratio of red to green fluorescence emission was significantly greater in the GFP-cA1-RFP clone compared to the GFP-cA1 only (GFP-cA1: $0.14 \pm 0.01$, $n = 6$; GFP-cA1-RFP: $0.41 \pm 0.04$, $n = 7$, $p = 0.0014$). To increase the sensitivity to local motions, we repeated the experiment with the RFP cloned into the C1 position (i.e. GFP-cA1-C1-RFP). Unfortunately, here again we did not detect any change in RFP/GFP ratio with either GFP-cA1-C1-RFP alone or when diluting this construct out with unlabeled wild-type cASIC1 ($\Delta F_{\text{RFP/GFP}}$ GFP-cA1-C1-RFP pH 6: $-0.9 \pm 1.4\%$, $n = 3$; + wild type pH 6: $-0.8 \pm 1.5\%$, $n = 5$; *Figure 4—figure supplement 1*). Although we did not detect any changes in FRET, we cannot exclude that extracellular acidification does cause the N and C tails do move apart laterally but such movements are either small, localized, or otherwise fall below our ability to detect them.

Finally, we investigated if either the N or C tails move axially, with respect to the plasma membrane, upon extracellular acidification. To do this, we returned to DPA quenching by first recording the fluorescence emission during a series of voltage steps in the absence of DPA, presence of DPA at pH 8.0, and then in the presence of DPA at pH 6.0. Importantly, the voltage-dependent quenching of Lck-CFP was not different between extracellular pH 8 compared to pH 6 ($\Delta F/F_{-15 \text{ mV}}$ pH 8: $0.45 \pm 0.03$, pH 6: $0.43 \pm 0.04$, $n = 6$, uncorrected p-value = 0.52, paired Wilcoxon signed-rank test, *Figure 5*), thus confirming the DPA quenching is relatively pH-tolerant. Interestingly, the extent of quenching at the N and C1 position was significantly reduced in pH 6 compared to pH 8 (*Figure 5*). Specifically, in the resting state (pH 8), CFP at the N or C1 position had a $\Delta F/F_{-15 \text{ mV}}$ of $0.49 \pm 0.02$ or $0.57 \pm 0.04$ ($n = 13$ or 11), respectively. However, with extracellular pH 6 (populating the desensitized state), these were reduced to $0.41 \pm 0.02$ and $0.55 \pm 0.03$. In a subset of cells expressing the N

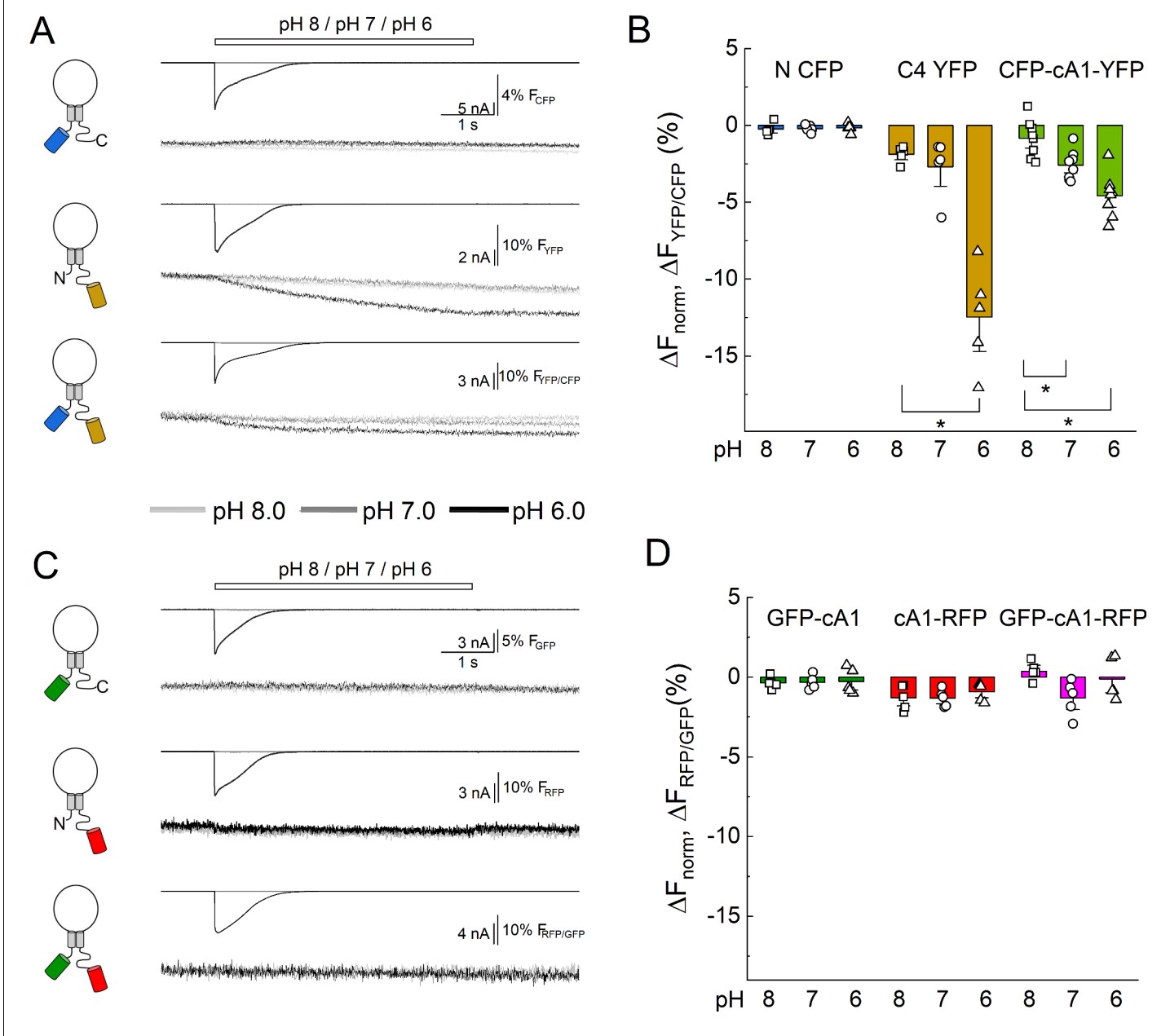

**Figure 4.** Extracellular acidification produces no change in fluorescence resonance energy transfer (FRET) when using pH-insensitive fluorescent proteins (FPs). (A) Cartoons (*left*), electrophysiology (*upper*), and fluorescence traces (*lower*) from single cells transfected with the CFP, YFP, or both FPs attached to cASIC1 during pH jumps from eight into pH 8, pH 7, or pH 6. Fluorescence trace is the acceptor fluorescence signal divided by the donor. (B) Summary of change in acceptor/donor ratio during pH changes. (C) Same as in A but for pH-insensitive variants of GFP and RFP (see Materials and methods). (D) Summary of change in acceptor/donor ratio during pH changes. Symbols indicate single cells (*N* = 5–8 cells), error bars are SEM. Asterisks denote statistical significance with p-values of 0.028 for YFP pH 8 versus 6, 0.031 and 0.0025 for CFP-cA1-YFP pH 8 versus 7 and pH 8 versus 6, respectively.

The online version of this article includes the following figure supplement(s) for figure 4:

**Figure supplement 1.** N and C terminal GFP and RFP do not change fluorescence resonance energy transfer (FRET) efficiency upon extracellular acidification.

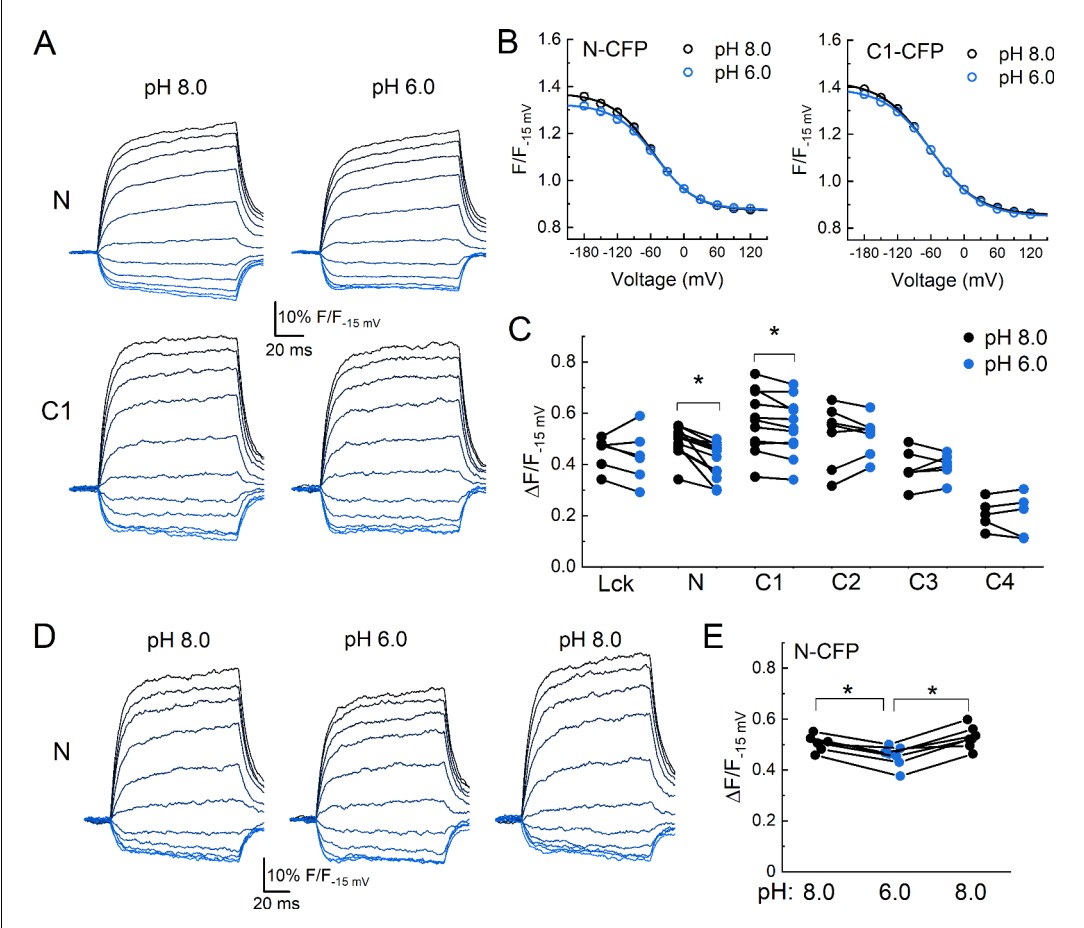

**Figure 5.** Acidification induces an axial motion of the membrane proximal region. (**A**) Example CFP fluorescence quenching traces from single cells expressing N or C1 CFP-cASIC1 with extracellular pH 8 (left) or pH 6 (right). (**B**) Quenching voltage curves for CFP inserted in N (left) or C1 (right) position. Curves in extracellular pH 8.0 (resting state) shown in black and pH 6.0 (desensitized) in blue. (**C**) Summary of the extents of quenching at each position with extracellular pH 8 (black) and pH 6 (blue). Symbols denote individual cells ($N$ = 5–13 cells per construct) and error bars represent SEM. Asterisks denote statistical significance by Wilcoxon signed-rank test after correction for multiple comparisons. The p-value for N and C1 are 0.0.0010 and 0.0.033, respectively. (**D**) Fluorescence-voltage traces from single cell expressing N CFP in the indicated extracellular buffer with 5 μM dipicrylamine (DPA). (**E**) Summary of the voltage-dependent extent of DPA quenching from N CFP in the indicated extracellular pH. Symbols denote individual cells ($N$ = 7) and error bars represent SEM. * indicates p-value < 0.05 by paired Wilcoxon signed-rank test.

The online version of this article includes the following figure supplement(s) for figure 5:

**Figure supplement 1.** Magnitude of $\Delta F/F$ change between pH 8.0 and 6.0 for CFP insertions.

CFP clone, we maintained the recording in pH 8, 6 and then back to pH 8 and found the effect on the extent of quenching was readily reversible (*Figure 5D–E*). None of the other positions showed significant evidence of movement between these conditions (*Figure 5C*, *Figure 5—figure supplement 1*), supporting that this motion is specific to the membrane proximal region and not a general pH artifact. These results demonstrate that both the N and C tails of cASIC1 move axially with respect to the membrane between the resting and desensitized states. To determine if the axial motion is toward or away from the plasma membrane, we considered our prior data and theoretical analysis (*Figure 2*, *Figure 2—figure supplement 5*). The resting state DPA quenching places the N terminal FP chromophore about 15–20 Å from the inner membrane leaflet (*Figure 2*, *Figure 2—figure supplement 5*). The reduced quenching in the pH 6 condition indicates the FP moves even closer to the plasma membrane upon channel desensitization. However, interpreting the C1 data is less straightforward. The DPA data indicate that FPs at the C1 position are about 23–25 Å from the inner leaflet, thus lying at or near the peak of CFP's $\Delta F/F_{norm}$ curve (*Figure 2—figure supplement 5C and D*). In such a position, moving either toward or away from the plasma membrane will result

in decreased quenching. A scenario where the N terminal moves toward the membrane while the C tail moves away is difficult to reconcile with the lack of ΔFRET between FPs in these positions upon acidification (*Figure 4*). Therefore, our data favor a model where the N and proximal segment of the C tail move up toward the inner membrane leaflet when the channel is exposed to desensitizing conditions (*Figure 6*).

## Discussion

Using a combination of patch clamp fluorometry and FLIM, we obtained an outline of the ASIC ICDs. We find the N terminus and proximal segment of the C terminus to extend about 6–10 and 12–16 Å into the cytosol, while the remainder of the C tail projects not more than 40 Å. Further, FLIM FRET results indicate the N and proximal segment of the C tail from the same subunit are physically closer than from different subunits, presumably reflecting association via the putative salt bridge. Finally, we find that the N and proximal segment of the C tail move toward the plasma membrane upon acidification, while we detect no substantial lateral movement (*Figure 6*). Our results support the emerging idea that extracellular acidification promotes ASIC ICD conformational changes, potentially creating or exposing sites within the membrane proximal segments that bind intracellular signaling partners and facilitating metabotropic signaling.

### Sources of error

The principle source of error is the possible disruption of the N or C tail caused by the addition or insertion of an FP. Insertion of an FP into the midst of a region with secondary structure would obviously distort this region. It is also possible that the addition of an FP would bias the region toward or away from the membrane. While we consider this unlikely given the solubility of FPs, we cannot exclude this possibility. Future studies could employ smaller fluorophores such as the fluorescent

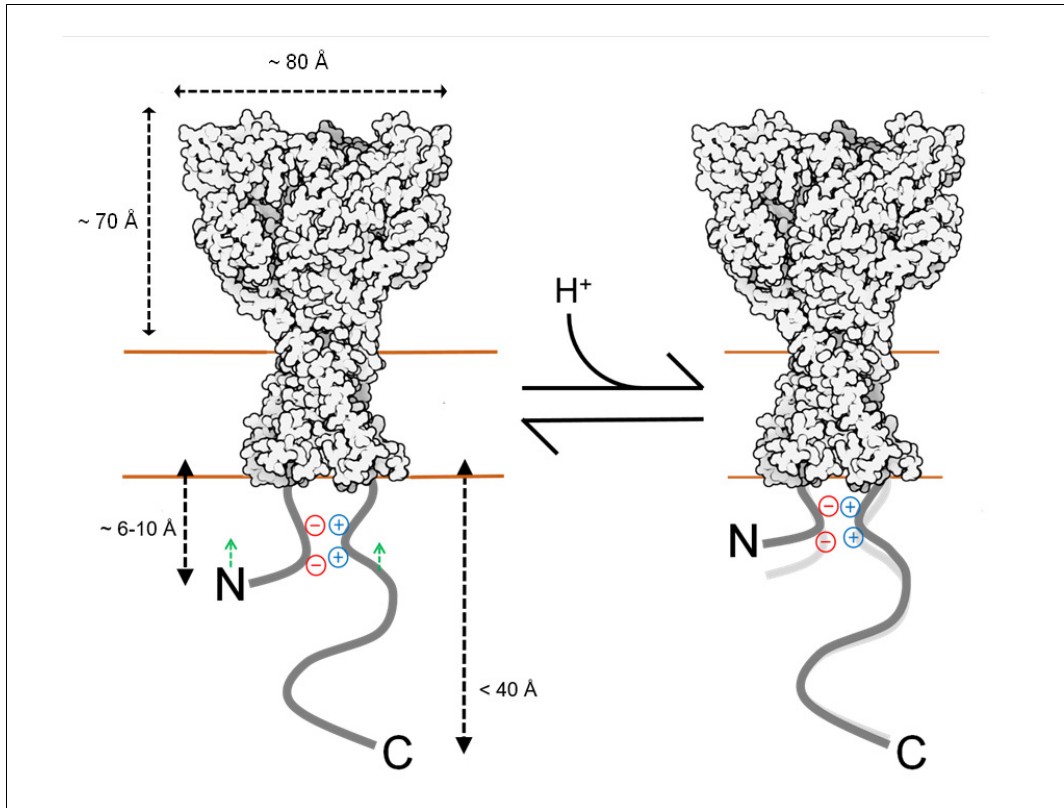

**Figure 6.** Cartoon of topography and acid-inducted motions for cASIC1. Estimated distances of N and C tail positions derived from dipicrylamine (DPA) quenching mapped onto resting state structure (6VTL). Not drawn to scale. Extracellular pH moves the proximal N and C segments toward the membrane.

non-canonical amino acid L-ANAP (*Chatterjee et al., 2013*; *Gordon et al., 2018*; *Gordon et al., 2016*; *Sakata et al., 2016*; *Zagotta et al., 2016*) or split inteins and semi-synthesis (*Khoo et al., 2020*; *Lueck et al., 2016*; *Shah and Muir, 2014*) to introduce small, bright fluorophores in specific sites. Such methods, ideally combined with acceptor groups (*Gordon et al., 2018*; *Gordon et al., 2016*; *Zagotta et al., 2016*), will refine and test the provisional map we report here. An additional source of error in our DPA quenching estimates is the background fluorescence from FPs in the ER (endoplasmic reticulum). These FPs may be either unquenched or progressively quenched over the course of the experiment as DPA migrates into the cell. In prior work, intracellular channels and FPs produced variable amounts of background fluorescence (*Barros et al., 2018*; *De-la-Rosa et al., 2013*; *Zachariassen et al., 2016*), which may confound more accurate mapping. Fortunately, our family of cASIC1 with FP insertions show very robust plasma membrane expression (*Figure 2—figure supplement 3*). We also selected cells with minimal intracellular fluorescence to reduce the impact of this background signal. In addition, our theoretical analysis treats DPA molecules as existing in two discrete planes, separated by 25 Å. In reality, DPA molecules will inhabit a distribution of distances centered around these planes which would introduce some inaccuracy in our measurements. A further source of potential error is the orientation dependence of FRET. This concern is partly mitigated since the fluorophores are attached to regions which are likely to be unstructured or highly flexible, allowing the FPs to adopt multiple orientations. Consistent with this, past theoretical and experimental work suggests a minimal influence of orientation dependence for FRET involving DPA as an acceptor (*Zachariassen et al., 2016*). Subsequent studies using isotropic resonance transfer methods such as lanthanide-based resonance energy transfer or transition-metal FRET may reduce many of these concerns and provide a more refined map (*Selvin, 2002*; *Taraska et al., 2009*).

Despite these sources of error, we do observe a consistent and reproducible decrease in the $\Delta F/F$ at the N and C1 position upon the addition of extracellular acidification in the same experiment (*Figure 5A–C*, *Figure 5—figure supplement 1*). This is not detected at other positions or with Lck-CFP and hence it does not appear to be a pH-induced artifact (*Figure 5C*, *Figure 5—figure supplement 1*). Indeed, in the subset of longer N position recordings, we were able to reverse this decrease in $\Delta F/F$ by moving back to extracellular pH 8 (*Figure 5D and E*).

## Structural interpretation

Structures of full-length cASIC1 have been solved using cryo-EM. The N and C tails are present in these constructs yet give no resolvable density (*Yoder and Gouaux, 2020*; *Yoder et al., 2018*). This suggests the tails maybe largely disordered, waiting to bind intracellular proteins via an induced fit or conformation selection mechanisms (*Arai et al., 2015*; *Oldfield and Dunker, 2014*). If true, then our data define the borders of the conformation landscape these disordered segments occupy. However, it is also possible that within these disordered tails are areas of secondary structure. A small but well-ordered segment of the tails which is moved around by adjacent disordered regions might be unresolvable by cryo-EM. Having isolated regions of order among a largely disordered tail is consistent with the proposed salt bridge between N and the C1 region. Secondary structure predictions using the JPred tool also support this view, with the proximal segment of the C tail having a strong tendency to form alpha helices while the remainder of the tail is unstructured (*Drozdetskiy et al., 2015*). Additional approaches such as CD spectroscopy combined with molecular modeling (*Ezerski et al., 2020*), NMR, and DEER will be critical to address the extent and location of secondary structure within the tails as well as further define the intracellular structural changes induced by extracellular pH that we report here. A particularly useful approach maybe single-molecule FRET, which has already been used to gain insight into the unstructured, and larger, intracellular tails of NMDA receptors (*Choi et al., 2013*; *Choi et al., 2011*).

## Physiological relevance

Numerous protein-protein interactions have been reported for ASICs that are important for channel trafficking, gating, and possibly metabotropic signaling (*Kellenberger and Schild, 2015*; *Zeng et al., 2014*). The best documented of these are PICK1, PSD-95, and stomatin (*Baron et al., 2002*; *Hruska-Hageman et al., 2004*; *Hruska-Hageman et al., 2002*; *Klipp et al., 2020*; *Leonard et al., 2003*; *Price et al., 2004*; *Zha et al., 2009*). Of particular note is the interaction

between ASIC3's C tail and stomatin, which enables stomatin to inhibit channel activation through a mechanism involving the first transmembrane helix (*Klipp et al., 2020*). In addition, recent work has suggested that the membrane proximal region of the ASIC1 C tail becomes liberated from the N tail during desensitization, allowing for the recruitment of RIP1 kinase (*Wang et al., 2020*). This results in activation of RIP1 by phosphorylation leading to eventual cell death via necroptosis. Canonical activation of RIP1 by the tumor necrosis factor (TNF) receptor 1 (TNFR1) occurs through heterodimerization of death domains (DD) present on both RIP1 and TNFR1 (*Meng et al., 2018*). ASIC1 does not contain a DD, so how RIP1 interacts with the C tail of ASIC1 and resulting in activation remains an important unanswered question. Using the FP insertion approach we adopted is problematic for investigating this, or any of the other protein-protein interactions mentioned above, due to the sheer disruptive size of the FP. However, using a higher resolution method with a smaller fluorophore could shed light on the molecular determinants, timing, and dynamics of these interactions.

# Materials and methods

## Key resources table

| Reagent type (species) or resource | Designation | Source or reference | Identifiers | Additional information |
|---|---|---|---|---|
| Cell line (*Homo sapiens*) | HEK293T ASIC1 KO | *Rook et al., 2020a* | Clone 2C1 | |
| Recombinant DNA reagent | pcDNA3.1-cocASIC1 | *Rook et al., 2020b* | | |
| Recombinant DNA reagent | pmVenus(L68V)-mTurquiose2 | | RRID:addgene_60493 | |
| Recombinant DNA reagent | Gamillus/pcDNA3 | | RRID:addgene_124837 | |
| Recombinant DNA reagent | PSD-95-pTagRFP | | RRID:addgene_52671 | |
| Recombinant DNA reagent | Lck-mScarlet-I | | RRID:addgene_98821 | |
| Commercial assay or kit | Q5 Hot Start High-Fidelity 2X Master Mix | New England Biolabs, Inc | M0494L | PCR |
| Commercial assay or kit | NEBuilder HiFi DNA Assembly Master Mix | New England Biolabs, Inc | E2621L | Insertion of FP tags |
| Chemical compound, drug | Polyethylenimine 25 k | Polysciences, Inc | 23966–1 | Transfection reagent |
| Chemical compound, drug | Dipicrylamine | Biotium | 60037 | Quencher |
| Software, algorithm | Axograph | | RRID:SCR_014284 | Patch clamp and fluorometry acquisition |
| Software, algorithm | Clampfit | Molecular Devices | RRID:SCR_011323 (pClamp) | Patch clamp and fluorometry analysis |
| Software, algorithm | OriginPro 2020b | OriginLab Corp | 9.7.5.184 (Student Version) | Data fitting and figure preparation |

## Plasmids and molecular cloning

A pcDNA3.1(+) vector containing the sequence for codon-optimized chicken ASIC1 (pcDNA3.1-cocASIC1) was used for all experiments (*Rook et al., 2020b*). Scar-less insertion of FP tags was achieved by opening the pcDNA3.1-cocASIC1 vector by PCR with primers containing 18-nt homologous overhangs. FP sequences were PCR-derived from pmVenus(L68V)-mTurquiose2 (Addgene plasmid #60493), Gamillus/pcDNA3 (Addgene plasmid #124837), or PSD-95-pTagRFP (Addgene #52671), and cloned in using NEBuilder HiFi DNA Assembly Master Mix (New England Biolabs) according to the manufacturer's instructions. These FP constructs were gifts from Dorus Gadella,

Takeharu Nagai, and Johannes Hell, respectively. All constructs were verified by Sanger sequencing (Eurofins Genomics). The Lck membrane tether clones were made using Lck-mScarlet-I (Addgene #98821), removing the mScarlet by inverse PCR and inserting PCR-derived mTurquiose2 or mVenus. Throughout the manuscript CFP, GFP, YFP, and RFP to refer to mTurquoise2, Gamillus, mVenus, and TagRFP, respectively.

## Cell culture and transfection

A clonal Human Embryonic Kidney 293T (HEK293T) cell line, purchased from ATCC and identity confirmed with STR profiling, with the endogenous ASIC1 gene knocked out were used for all experiments (*Rook et al., 2020a*). A PCR test for mycoplasma contamination, last performed on 6/04/2021, was negative. HEK293T cells were maintained in minimum essential medium containing glutamine and Earle's salts (Gibco) supplemented with 10% fetal bovine serum (Atlas Biologicals) and PenStrep (Gibco). Cells were plated in 35 mm tissue culture treated dishes and transfected 1–2 days later using polyethylenimine 25 k (Polysciences, Inc) with a mass ratio of 1:3 (cDNA:PEI). For dipicrylamine (DPA) quenching experiments, 500 ng of pcDNA3.1-cocASIC1 with an FP tag and 2.0 µg of pcDNA3.1(+)-empty vector were co-transfected per 35 mm dish. We found that dilution with the empty vector helped limit intracellular accumulation of the channel. Cells were dissociated 24–48 hr post-transfection with divalent-free DPBS (Gibco) supplemented with 0.5 mM EDTA and sparsely seeded onto 18 mm No. 1 coverslips that had been treated with 100 µg/mL poly-lysine for 20 min.

## Electrophysiology

For the experiments in *Figure 2—figure supplement 2*, excised patch recordings were done as described previously (*Rook et al., 2020b*). Briefly, borosilicate patch pipettes were pulled and heat-polished to a resistance of 3–6 MΩ. The internal pipette solution was (in mM) 135 CsF, 11 EGTA, 10 HEPES, 10 MES, 2 $MgCl_2$, 1 $CaCl_2$, and pH adjusted to 7.4 using CsOH. External solutions were comprised of (in mM) 150 NaCl, 1 $CaCl_2$, 1 $MgCl_2$, and either 20 HEPES (pH 7.0 or 8.0) or 20 MES (pH 5.0 or 6.0) and adjusted using NaOH. Data were acquired at 20–50 kHz and filtered online at 10 kHz using AxoGraph software (Axograph), an Axopatch 200B amplifier (Molecular Devices) and USB-6343 DAQ (National Instruments) at room temperature and with a holding potential of −60 mV. Series resistance was routinely compensated by 90–95% when the peak amplitude exceeded 100 pA. A homebuilt double or triple barrel perfusion pipette (*MacLean, 2015*) (Vitrocom) attached to a piezo translator (Physik Instrumente) under computer control was used for fast perfusion. Piezo voltage commands were generally filtered between 50 and 100 Hz.

## Patch clamp fluorometry

Coverslips were visualized using an S Fluor 40× oil-immersion objective (1.30 NA, Nikon) mounted on a Nikon Ti2 microscope. Transfected cells were excited and visualized with either a four-channel LED or single-channel LED (Thorlabs) and dichroic filter cube. Single cells were patched in a whole-cell configuration using heat-polished borosilicate glass pipettes with 3–6 MΩ resistance. Cells were continually perfused with external solution, with or without 5 µM DPA (Biotium), using a homebuilt multi-barrel flow pipe (see above). Single cell fluorescence was collected using a D-104 dual-channel photometer (Photon Technology International) attached to the left microscope port. Emitted fluorescence was limited to only the patched cell using adjustable slits and detected with photomultiplier tubes (Hamamatsu Photonics, R12829). Photometers were generally set to a voltage of 650–900 V and filtered with a time constant of 0.5 ms. The photometer voltage signals were fed to the analog inputs of the same DAQ as the patch clamp amplifier and recorded using AxoGraph. The timing of the excitation source was triggered by TTL.

For DPA quenching experiments with CFP, a 455 nm LED, 455/20 nm excitation filter, 442 nm dichroic, and 510/84 nm emission filter were used. For YFP, a 505 nm LED, 510/20 excitation filter, 525 dichroic, and 545/35 band-pass emission filter were used. In all cases, we worked to minimize background fluorescence by selecting cells with prominent plasma membrane fluorescence and minimal intracellular signal. Quenching curves were generated in whole-cell configuration by a series of 100 ms steps ranging from −180 to +120 mV in 30 mV increments, with −15 mV holding potential and 1 s between steps. Excitation was triggered 50 ms before a voltage step and lasted for 200 ms. This family of voltage steps was repeated four times either in control, with continual perfusion of 5

µM DPA, or during DPA washes with several seconds separating each condition. We found the data from the first family of DPA steps tended to be inconsistent, presumably reflecting DPA equilibration within the cell. Thus, only the final three series of steps were averaged together. For each recording a background signal was measured by slightly moving the stage to a region with no cells and this signal subtracted. Analysis was done using Clampfit (Molecular Devices) where, following background subtraction, the average fluorescence signal in the second quintile (20 ms) window within the 100 ms voltage step was divided by the fluorescence signal at the preceding holding potential to generate $F/F_{-15mV}$. These values were plotted in OriginLab (OriginLab Corp) and cells were individually fitted with a Boltzmann function. The difference of the minimum and maximum values of the Boltzmann gave the $\Delta F/F_{-15mv}$ for that specific cell.

For patch clamp FRET experiments, the fluorescence emission was split into two photomultiplier tubes. For CFP/YFP, a 455 nm LED, 455/20 nm excitation, 442 nm dichroic, and 510/84 nm wideband emission was used. This combination of donor and acceptor emission was passed to the dual photometer, through the adjustable aperture, and split by a 518 nm dichroic before further filtering into nominal donor (482/25 nm) and acceptor (537/26 nm) emission channels. A similar configuration was used for GFP/RFP with 455 nm LED, 470/10 nm excitation, 495 nm dichroic, 500 nm long-pass emission followed by a 552 nm splitter diving donor (510/42 nm) and acceptor (609/54 nm). Single cells were patch-clamped and piezo-driven perfusion used to switch between pH 8.0, pH 7 and 6.0 for 2 s, with 20 s between acidic applications. The excitation source was triggered 1 s before the solution switch for 10 s. The background subtracted 'acceptor' channel fluorescence signal was divided by the subtracted 'donor' channel signal to yield an apparent FRET signal. The goal of this experiment was to ascertain if any changes in FRET are detectable upon activation/desensitization of the channel, not to quantify the FRET efficiency. Therefore, we made no attempt to correct for bleedthrough excitation or emission. To reflect this, all such measurements are referred to as $\Delta FRET_{app}$ or change in apparent FRET.

## FLIM and confocal imaging

FLIM was done as described previously (*McCullock et al., 2020*). Cells were transfected in either 35 or 60 mm culture dishes and imaged using a water immersion 25× objective (XL Plan N, 1.05 NA) mounted on an Olympus IX61WI upright microscope. A Mai Tai Ti:Sapphire multi-photon laser (Spectra Physics) was used for excitation with an 860 nm wavelength, a repetition rate of 80 MHz and pulse width of approximately 100 fs. Donor emission was filtered by a 480–20 filter and measured by a H72422P Hamamatsu hybrid avalanche photodiode. Time-correlated single photon counting was done using a Becker and Hickl card with a resolution of 25 ps. Using VistaVision software (ISS), donor fluorescence from the plasma membrane from individual cells was binned and fit with a single exponential function, consistent with the lifetime of CFP variant mTurquoise2 (*Goedhart et al., 2012*).

For confocal imaging, cells were transfected with Lck-CFP or CFP-tagged cASIC1 in 35 mm dishes. After 2 days, cells were stained with 2 mL of 7.5 µM FM1–43 (Invitrogen) immediately prior to imaging with an Olympus FV1000MP microscope using a 60× water immersion objective (U Plan SApo, 1.20 NA). CFP and FM1–43 were simultaneously excited with a 440 nm laser and emissions between 465 and 495 nm collected as CFP and 575 and 675 nm collected as FM1–43.

## DPA quenching simulations

Closely following the work of *Wang et al., 2010* and *Zachariassen et al., 2016*, we calculated the FRET between a donor fluorophore and a plane of DPA molecules over a range of axial distances. The donor fluorophore was positioned at a distance, $R_a$, from a point within a perpendicular plane representing the inner leaflet of the plasma membrane. This point is the center of a ring of radius $r$, within the perpendicular plane. The distance between the donor and DPA on the perimeter of the ring is the hypotenuse of a triangle with the other sides being the ring radius, $r$, and the axial distance between the center of the ring and the donor, $R_a$. Thus the donor-acceptor distance is $\sqrt{R_a^2 + r^2}$. We define $P(r)$ as the probability that no FRET will occur between the donor and DPA positioned along ring perimeter and $Q(r)$ as the probability FRET (quenching) will occur. As the sum of $P(r)$ and $Q(r)$ is one, for a ring of slightly larger radius $r + dr$:

$$P(r + dr) = P(r)(1 - Q(r)dr)$$

FRET efficiency, $E$, over a distance, $d$, for a single donor-acceptor pair is given by:

$$E = \frac{1}{1 + \left(\frac{d}{R_0}\right)^6}$$

where $R_0$ is the Förster distance giving half-maximal FRET. As with past work, we used 5 µM DPA which gives a density, $\sigma$, of $1.25 \times 10^{-4}$ molecules per Å$^2$ (*Wang et al., 2010*). Combing the DPA density and distance constraint, we define the probability of quenching as:

$$Q(r)dr = \frac{\sigma 2\pi r \, dr}{1 + \left(\frac{R_a^2 + r^2}{R_0^2}\right)^3}$$

Given *Equation 1*, the following differential equation can be derived:

$$\frac{dP}{dr} = \frac{\sigma 2\pi r P}{1 + \left(\frac{R_a^2 + r^2}{R_0^2}\right)^3}$$

Substituting $u = \left(R_a^2 + r^2\right)/R_0^2$ and evaluating the integral at $r = \infty$ gives:

$$P_\infty = exp\left(-\sigma\pi R_0^2 \int_{\frac{R_a^2}{R_0^2}}^{\infty} \frac{du}{1 + u^3}\right)$$

which represents the unquenched fluorescence signal remaining over a range of axial distances. As DPA is assumed to traverse about 25 Å in the plasma membrane (*Wang et al., 2010*; *Zachariassen et al., 2016*), evaluating this integral using either $R_a$ or $R_a + 25$ gives the quenching curves for DPA at the inner leaflet or outer leaflet, which are populated by infinitely depolarizing or hyperpolarizing conditions, respectively. $R_0$ calculations were done with the $k^2 = 2/3$ assumption, using donor FP spectra and quantum yield listed in FP database (*Lambert, 2019*). The DPA absorption spectrum was measured with 25 µM DPA diluted in saline on a QuickDrop spectrophotometer (Molecular Devices). The DPA molar extinction coefficient at 420 nm was determined to be 26,500 M$^{-1}$cm$^{-1}$ by fitting the slope of absorption measurements from serial dilutions of DPA into saline or 1% SDS. This value is consistent with previously reported measurements (*Gunupuru et al., 2014*).

### Statistics and data analysis

Current desensitization decays were fitted using exponential decay functions in Clampfit (Molecular Devices). For recovery from desensitization experiments, the test peak (i.e. the second response) was normalized to the conditioning peak (i.e. the first response). OriginLab (OriginLab Corp) was used to fit the normalized responses to:

$$I_t = \left(1 - e^{\left(\frac{-t}{\tau}\right)}\right)^m \tag{1}$$

where $I_t$ is the fraction of the test peak at an interpulse interval of $t$ compared to the conditioning peak, $\tau$ is the time constant of recovery, and $m$ is the slope of the recovery curve. Patches were individually fit and averages for the fits were reported in the text. $N$ was taken to be a single patch.

FRET$_{efficiency}$ from FLIM measurements was calculated using:

$$FRET_{efficiency} = 1 - \left(\frac{\tau_{donor + acceptor}}{\tau_{donor \, only}}\right)$$

where $\tau_{donor \, only}$ and $\tau_{donor+acceptor}$ are the single exponential time constants of donor decay in the absence and presence of acceptor, respectively.

Unless otherwise noted, statistical testing was done using nonparametric permutation or randomization tests with at least 100,000 iterations implemented in Python to assess statistical significance.

Reported p-values have been adjusted using the Holm-Bonferroni method for multiple test corrections. Data for each figure can be found in *Source data 1*.

## Acknowledgements

We thank Dr Nicolas Masse, California Institute of Technology/University of Chicago, for help with DPA quenching calculations, Lyndee Knowlton for her technical assistance, and Dr David Yule for generous sharing of equipment. This work was funded by NIH R35GM127951 to DMM, and NSF Graduate Research Fellowship to TC. TC and TWM were supported in part by the University of Rochester Harold C Hodge Memorial Fund.

## Additional information

### Funding

| Funder | Grant reference number | Author |
|---|---|---|
| National Institute of General Medical Sciences | R35GM137951 | David M Maclean |
| National Science Foundation | | Tyler Couch |
| University of Rochester | | Tyler Couch<br>Tyler W McCullock |

The funders had no role in study design, data collection and interpretation, or the decision to submit the work for publication.

### Author contributions

Tyler Couch, Formal analysis, Funding acquisition, Investigation, Visualization, Writing - review and editing; Kyle D Berger, Formal analysis, Investigation, Visualization, Writing - review and editing; Dana L Kneisley, Formal analysis, Investigation, Writing - review and editing; Tyler W McCullock, Resources, Investigation, Writing - review and editing; Paul Kammermeier, Supervision, Funding acquisition, Writing - review and editing; David M Maclean, Conceptualization, Formal analysis, Supervision, Funding acquisition, Investigation, Visualization, Writing - original draft, Writing - review and editing

### Author ORCIDs

Tyler W McCullock ⬤ http://orcid.org/0000-0003-1628-1102
David M Maclean ⬤ https://orcid.org/0000-0001-8294-6075

### Decision letter and Author response

Decision letter https://doi.org/10.7554/eLife.68955.sa1
Author response https://doi.org/10.7554/eLife.68955.sa2

## Additional files

### Supplementary files

- Source data 1. Patch clamp and fluorescence experimental data.
- Transparent reporting form

### Data availability

All analyzed results contributing to this study are included in the manuscript and supporting files. Source data files have been provided for all figures containing data.

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
