## [Decision Letter]

**Acceptance summary:**

This is a rigorous and clearly written paper that provides quantitative data showing small, previously uncharacterized, movements of intracellular regions of ASIC channels. These channels are involved in pain signalling and other processes, and apparently can couple to intracellular pathways independent of ion flow. Here the authors measure the organization and movements of the unstructured intracellular parts of ASIC using fluorescence spectroscopy coupled to functional measurements. These careful experiments might provide a basis to understand signaling mediated by these ion channels. The authors have carried out the requested changes and satisfactorily answered reviewer's comments.

**Decision letter after peer review:**

Thank you for submitting your article "Topography and motion of the acid-sensing ion channel intracellular domains" for consideration by *eLife*. Your article has been reviewed by 3 peer reviewers, including Leon D Islas as the Reviewing Editor and Reviewer #1, and the evaluation has been overseen by Kenton Swartz as the Senior Editor. The following individuals involved in review of your submission have agreed to reveal their identity: Andrew J R Plested (Reviewer #2); Michael C Puljung (Reviewer #3).

Essential revisions:

1) The measurement of axial movement in response to pH changes yielded very small distances. This makes the interpretation of these data complicated. The authors should discuss all possible sources of error involved in these experiments and thoroughly discuss this particular result in the light of possible artifacts.

2) The authors should discuss the appropriateness of statistical methods used in some sections of the paper and again, discuss results and conclusions reached in the light of sources of error. These possible errors are pointed out in detail by the reviewers below and should be addressed thoroughly.

*Reviewer #1 (Recommendations for the authors):*

This manuscript by Couch et al. makes use of different fluorescence spectroscopy techniques as approximations to provide a coarse definition of the organization and possible orientations of the N- and C-terminal domains of the ASIC channel, which are intracellular. The experiments are nicely executed and controlled and the general conclusion reached by the authors seem to be supported by the data.

However, the last section of the paper, which is an attempt at detecting movements or conformational changes of these regions is much less solid.

The main problem I see is that the measured changes in fluorescence produced by DPA quenching of the FPs are buried in the variability of the experiments. Although for position C1 it seems the statistical analysis detects a difference in quenching between ph 8 and 6, this effect is really small and seems to be produced by two or three patches.

This is likely due to the large size of the PF being used here and other factors that are not being taken into account.

So, the data presented in figure 5 has very little statistical power and its interpretation is difficult. This data should be removed or presented as a negative result.

A possible source of error or artifact, not discussed by the authors, is the possibility that the insertion of the large FPs actually distorts the real position/orientation of the terminal domains, especially the C-terminus. It should be taken into account that it is entirely possible that the C-terminus is parallel to the membrane and is being "dragged down" by the massive FPs. The authors should explain or justify why they think this is unlikely, and that their results are actual mirrors of reality.

One of the problems when using DPA and interpreting quenching with simulations is that the authors assume that all DPA molecules at a given voltage, even extreme voltages, exist in a single plane within the membrane. However, statistical mechanics says that DPA should exist in a Boltzmann distribution of distances. So, quenching of an FP will occur from such a distribution of distances and not only from distances obtained from the radial distribution of molecules in a single plane. This results in a smear or broadening of the voltage-dependence of the quenching function and an increased error at the edges of these distributions.

*Reviewer #2 (Recommendations for the authors):*

This paper is a model of clarity and rigor. It was a pleasure to read. I have no suggestions for improvement. The dilution experiments in Figure 3 are very good indeed. I congratulate the authors on their admirable work.

*Reviewer #3 (Recommendations for the authors):*

– Can concatemeric channels like CFP-cA1-cA1 traffic to the plasma membrane when expressed alone? Can they form functional channels (for example with two subunits from one concatemer and one subunit from a different concatemer)?

– The use of the YFP to CFP ratio to quantify FRET is not standard. The authors rightly point out that they do not use this metric quantitate results. However, referring to it as FRETapp may be confusing to the reader. The authors should simply refer to it as FYFP/FGFP in figure 4 A and C and ΔFYFP/FGFP in Figure 4 B and D (and the associated text).

– The authors should specify which 20 ms window was used to quantify the fluorescence changes in Figures 1, 2, and 5.

– In providing approximate locations relative to the plasma membrane for their fluorescent protein (FP) tags, the authors add 10 Å to the axial distance to approximate the diameter of the β-barrel structure of their tags. Presumably this assumes that the long axis of the FP is parallel to the plasma membrane. How would this adjustment change if the barrel were oriented perpendicular to the membrane?

– Whereas the authors do explain ΔF/F-15 mV in the methods section, they may wish to do so in the main text on page 15 for the casual reader who may not have read the methods.

– The authors should provide explanation of terms like "thumb domain" on page 18 as a courtesy to their broader, non-ASIC-savvy audience.

– The authors should spell out dipicrylamine (DPA) the first time it is used on page 7. The word is spelled out in full on the following page, but without the abbreviation.

– On page 22, it is stated that YFP is pH sensitive because it is derived from GFP, which is also pH sensitive. This statement should be rephrased as it indirectly implies that the other tags (e.g. CFP) are not derived from GFP.

– It would be useful to the reader if the authors were to provide a plot of the calculated distance dependence of FRET between the CFP/YFP and GFP/RFP tags being used.

– Several citations were included for the use of the amino acid ANAP as a fluorescent label. The authors might consider replacing these with references to Chatterjee et al. (2013). Journal of the American Chemical Society, 135, 12540-12543. (often cited for ANAP expression) or Kalstrup and Blunck (2013). PNAS, 110:8272-8277. (The first use of ANAP as an ion channel probe).

– Similarly, the authors mention LRET and tmFRET. These abbreviations should be spelled out and appropriate literature citations added.

– The authors also use the abbreviation smFRET for single-molecule FRET. This can be spelled out as well, for the non-specialist audience.

– In Figure 2—figure supplement 2A, the authors should indicate on their plot where the change was made from pH 5 to pH 8.

– Figure 5—figure supplement 1 makes an important point. The authors should consider moving it to the main body of the paper.

---

## [Author Response]

Essential revisions:1) The measurement of axial movement in response to pH changes yielded very small distances. This makes the interpretation of these data complicated. The authors should discuss all possible sources of error involved in these experiments and thoroughly discuss this particular result in the light of possible artifacts.

We thank the reviewers and editors for their suggestions on this point. In our initial submission we described three principle sources of error using FP donors with DPA: disruption caused by the insertion of FP, background unquenched fluorescence and orientation-dependence (lines 442-463 of original manuscript). We have added or modified this list in several ways.

First, we have added the point by R2 that DPA is never 100% in one leaflet or another. Given the non-linear distance dependence of FRET, a small fraction of DPA in the ‘wrong’ leaflet (ie. inner leaflet DPA during hyperpolarization) would reduce our measured delF (lines 481-484 in new manuscript).

Second, we have expanded upon the impact of unquenched (or partially quenched) FPs within the cell. As we worked on implementing this method, we found the largest source of variance to be levels of intracellular FP from cell to cell. To reduce intracellular accumulation of FP-tagged channels, we found it very helpful to limit the expression by co-transfecting pcDNA3.1-empty vector at a 1:4 mass ratio (described in Cell culture and transfection). As also outlined in the methods, we selected cells with minimal intracellular FPs where possible. Despite these efforts, some imprecision remains.

To increase our confidence in these measurements, we used a repeated measures design, determining quenching first at pH 8 then at 6 and, in a subset of cells for N position, back at pH 8 again. To emphasize this, we have moved the data for the cells going the pH 8-6-8 from a supplemental figure to a new panel in Figure 5, as suggested by R3. Furthermore, following a suggestion from R2, we show the change in ΔF/F (ie. ΔΔF/F) when moving from pH 6 to pH 8 in a new supplemental figure (Figure 5—figure supplement 1). As you can see, all experiments at the N position and all but one cell for C1 show a decrease in quenching between pH 8 and 6. However, in the Lck, C2, C3 and C4 positions, some cells go up and others go down. Thus while the actual axial displacements we report may suffer from the sources of error above, we think our data do constitute reasonable evidence that some motion, however small, does occur at the N and C1 positions. We have added a condensed version of the above to our conclusions.

2) The authors should discuss the appropriateness of statistical methods used in some sections of the paper and again, discuss results and conclusions reached in the light of sources of error. These possible errors are pointed out in detail by the reviewers below and should be addressed thoroughly.

Our thanks to all for their attention and insight into these issues. Over the last few years, our lab has gravitated towards using randomization/permutation tests for significance since these tests make no assumption of normality (often difficult to confidently assess with typical N values), can provide an exact p value and are simply a much more intuitive way of detecting significance. However, in our initial submission we did not correct for multiple comparisons. In our revised submission, we have used the Holm-Bonferroni method to correct for multiple comparisons for all our randomization or Wilcoxon tests within any given panel.

For the paired or repeated measures data in Figure 5, in our initial submission we used a Wilcoxon *rank sum* test (non-parametric two sample aka Mann Whitney U). We used this test in error. We intended to use a Wilcoxon signed-rank test (non-parametric paired sample). There are two reasons for this choice. First, these experiments are done in the same cell and hence paired. Second, it so happens that in this case the N ΔF/F pH 8 data is not normally distributed anyways (p = 0.028 for Shapiro-Wilk). Therefore, the paired Wilcoxon signed-rank test is an appropriate choice. To be consistent, we used this test for all comparisons in this figure even though other tests pass normality checks. In our revised submission, we have used the paired Wilcoxon signed-rank test and corrected for the multiple comparisons (Holm-Bonferroni). We apologize for this mistake.

Reviewer #1 (Recommendations for the authors):This manuscript by Couch et al. makes use of different fluorescence spectroscopy techniques as approximations to provide a coarse definition of the organization and possible orientations of the N- and C-terminal domains of the ASIC channel, which are intracellular. The experiments are nicely executed and controlled and the general conclusion reached by the authors seem to be supported by the data.However, the last section of the paper, which is an attempt at detecting movements or conformational changes of these regions is much less solid.The main problem I see is that the measured changes in fluorescence produced by DPA quenching of the FPs are buried in the variability of the experiments. Although for position C1 it seems the statistical analysis detects a difference in quenching between ph 8 and 6, this effect is really small and seems to be produced by two or three patches.This is likely due to the large size of the PF being used here and other factors that are not being taken into account.So, the data presented in figure 5 has very little statistical power and its interpretation is difficult. This data should be removed or presented as a negative result.

Thank you for your comment. As other reviewers shared similar concerns, we wrote a combined response as part of essential revision 1. In answer to these specific points, as seen in our new Figure 5—figure supplement 1, every single cell showed less ΔF/F at the N position upon extracellular acidification (n = 13). At C1, all but one cell went down (n = 11). The other positions have some go up and some go down. The absolute value of the change maybe small, but it is consistent and reproducible at the N and C1 position and is not observed elsewhere.

A possible source of error or artifact, not discussed by the authors, is the possibility that the insertion of the large FPs actually distorts the real position/orientation of the terminal domains, especially the C-terminus. It should be taken into account that it is entirely possible that the C-terminus is parallel to the membrane and is being "dragged down" by the massive FPs. The authors should explain or justify why they think this is unlikely, and that their results are actual mirrors of reality.

In line 442 of our submission, we note that the insertion or addition of FPs maybe disruptive. We were thinking of how an insertion of FP would disrupt any local secondary structure around the insertion site. As far as FPs ‘dragging down’ N or C termini, we do not consider this likely and it is not clear to us what force would be ‘pulling’ on the FPs ‘down’. FPs are soluble, neither rise to the top of an aqueous solution nor precipitate at 1 x g. Moreover, expression of FPs alone results in diffuse cytosolic distribution. Thus, while we cannot rule out such an effect, we do not consider it especially likely. We have amended the text to read:

“The principle source of error is the possible disruption of the N or C tail caused by the addition or insertion of a fluorescent protein (FP). […] While we consider this unlikely given the solubility of FPs, we cannot exclude this possibility.”

One of the problems when using DPA and interpreting quenching with simulations is that the authors assume that all DPA molecules at a given voltage, even extreme voltages, exist in a single plane within the membrane. However, statistical mechanics says that DPA should exist in a Boltzmann distribution of distances. So, quenching of an FP will occur from such a distribution of distances and not only from distances obtained from the radial distribution of molecules in a single plane. This results in a smear or broadening of the voltage-dependence of the quenching function and an increased error at the edges of these distributions.

Thank you for pointing this out. We have added the following comment:

“In addition, our theoretical analysis treats DPA molecules as existing in two discrete planes, separated by 25 angstroms. In reality, DPA molecules will inhabit a distribution of distances centered around these planes which would introduce some inaccuracy in our measurements.”

Reviewer #3 (Recommendations for the authors):– Can concatemeric channels like CFP-cA1-cA1 traffic to the plasma membrane when expressed alone? Can they form functional channels (for example with two subunits from one concatemer and one subunit from a different concatemer)?

Remarkably, concatenated dimers (and tetramers) do make it to the plasma membrane and give wild-type like currents, presumably by joining up with other un-partnered subunits to form trimers. To our knowledge, this was first reported in van Bremmelen et al., PLoS 2015 (see PMID 26252376, Figure 5). In our case, because we co-express a wild type monomer we expect that such ‘subunit sharing’ is minimal.

– The use of the YFP to CFP ratio to quantify FRET is not standard. The authors rightly point out that they do not use this metric quantitate results. However, referring to it as FRETapp may be confusing to the reader. The authors should simply refer to it as FYFP/FGFP in figure 4 A and C and ΔFYFP/FGFP in Figure 4 B and D (and the associated text).

We thank the review for this suggestion. We have changed the figures and text as suggested.

– The authors should specify which 20 ms window was used to quantify the fluorescence changes in Figures 1, 2, and 5.

Thank you. We modified the following text to add clarity:

“Analysis was done using Clampfit (Molecular Devices) where, following background subtraction, the average fluorescence signal in the second quintile (20 ms) window within the 100 ms voltage step was divided by the fluorescence signal at the preceding holding potential to generate F/F_-15mV_.”

– In providing approximate locations relative to the plasma membrane for their fluorescent protein (FP) tags, the authors add 10 Å to the axial distance to approximate the diameter of the β-barrel structure of their tags. Presumably this assumes that the long axis of the FP is parallel to the plasma membrane. How would this adjustment change if the barrel were oriented perpendicular to the membrane?

The FP barrel has a radius of ~10 angstroms and a length of ~40 angstroms. Using the ‘perpendicular orientation assumption’ mentioned above, and the resulting 20 angstrom offset, would suggest that FPs preferentially orient in that way. We have no reason to think that is the case. Instead, we suspect the FPs are somewhat flexible and have relatively random orientations. Therefore, using the shortest distance seemed, to us, the best approximation.

– Whereas the authors do explain ΔF/F-15 mV in the methods section, they may wish to do so in the main text on page 15 for the casual reader who may not have read the methods.

Thank you for the suggestion. We now include:

“… and was quantified as the difference between the top and bottom values of a Boltzmann sigmoidal fit of the normalized fluorescence.”

– The authors should provide explanation of terms like "thumb domain" on page 18 as a courtesy to their broader, non-ASIC-savvy audience.

In lieu of a text explanation we have added a new supplemental figure to figure 2 showing this distance.

– The authors should spell out dipicrylamine (DPA) the first time it is used on page 7. The word is spelled out in full on the following page, but without the abbreviation.

This has been corrected, thank you.

– On page 22, it is stated that YFP is pH sensitive because it is derived from GFP, which is also pH sensitive. This statement should be rephrased as it indirectly implies that the other tags (e.g. CFP) are not derived from GFP.

Thank you. This sentence has been rewritten as “The chromophores of GFP and YFP are intrinsically pH sensitive making them unsuitable for studying acid-evoked conformational changes.”

– It would be useful to the reader if the authors were to provide a plot of the calculated distance dependence of FRET between the CFP/YFP and GFP/RFP tags being used.

We have elected not to do this since we feel it encourages the practice of converting FRET efficiency measured from FPs directly into a distance without due consideration for orientation dependence or other factors.

– Several citations were included for the use of the amino acid ANAP as a fluorescent label. The authors might consider replacing these with references to Chatterjee et al. (2013). Journal of the American Chemical Society, 135, 12540-12543. (often cited for ANAP expression) or Kalstrup and Blunck (2013). PNAS, 110:8272-8277. (the first use of ANAP as an ion channel probe).

We have added in the Chatterjee and Kalstrup references. We also added another citation from the Okamura lab where Anap was used in conjunction with DPA. However, we have retained our original citations of the Gordon-Zagotta studies.

- Similarly, the authors mention LRET and tmFRET. These abbreviations should be spelled out and appropriate literature citations added.

Thanks for catching this. We have corrected and now cite Selvin 2002 for LRET and Taraska 2009 PNAS for tmFRET.

– The authors also use the abbreviation smFRET for single-molecule FRET. This can be spelled out as well, for the non-specialist audience.

This has been corrected, thank you.

– In Figure 2—figure supplement 2A, the authors should indicate on their plot where the change was made from pH 5 to pH 8.

We have added this into the figure now.

– Figure 5—figure supplement 1 makes an important point. The authors should consider moving it to the main body of the paper.

Thank you very much for this excellent suggestion! We have done so and it is now in main Figure 5.